# Structural and molecular insight into the pH-induced low-permeability of the voltage-gated potassium channel Kv1.2 through dewetting of the water cavity

Juhwan Lee[1,2,3], Mooseok Kang[1,4], Sangyeol Kim[1,5]*, Iksoo Chang[1,3,4,5]*

**1** Center for Proteome Biophysics, DGIST, Daegu, Korea, **2** Department of Emerging Material Sciences, DGIST, Daegu, Korea, **3** Core Protein Resources Center, DGIST, Daegu, Korea, **4** Supercomputing Bigdata Center, DGIST, Daegu, Korea, **5** Department of Brain and Cognitive Sciences, DGIST, Daegu, Korea

* sykim@dgist.ac.kr (SK); iksoochang@dgist.ac.kr (IC)

**Data Availability Statement:** Deposit files We performed MD simulations using GROMACS 5.0.6 that can be obtained from http://www.gromacs.org/Downloads_of_outdated_releases We provide

## Abstract

Understanding the gating mechanism of ion channel proteins is key to understanding the regulation of cell signaling through these channels. Channel opening and closing are regulated by diverse environmental factors that include temperature, electrical voltage across the channel, and proton concentration. Low permeability in voltage-gated potassium ion channels (Kv) is intimately correlated with the prolonged action potential duration observed in many acidosis diseases. The Kv channels consist of voltage-sensing domains (S1–S4 helices) and central pore domains (S5–S6 helices) that include a selectivity filter and water-filled cavity. The voltage-sensing domain is responsible for the voltage-gating of Kv channels. While the low permeability of Kv channels to potassium ion is highly correlated with the cellular proton concentration, it is unclear how an intracellular acidic condition drives their closure, which may indicate an additional pH-dependent gating mechanism of the Kv family. Here, we show that two residues E327 and H418 in the proximity of the water cavity of Kv1.2 play crucial roles as a pH switch. In addition, we present a structural and molecular concept of the pH-dependent gating of Kv1.2 in atomic detail, showing that the protonation of E327 and H418 disrupts the electrostatic balance around the S6 helices, which leads to a straightening transition in the shape of their axes and causes dewetting of the water-filled cavity and closure of the channel. Our work offers a conceptual advancement to the regulation of the pH-dependent gating of various voltage-gated ion channels and their related biological functions.

## Author summary

The acid sensing ion channels are a biological machinery for maintaining the cell functional under the acidic or basic cellular environment. Understanding the pH-dependent gating mechanism of such channels provides the structural insight to design the molecular strategy in regulating the acidosis. Here, we studied the voltage-gated potassium ion

all input files to run the MD simulations including input coordinates, topologies and parameter files and etc. Additionally, we provide all the trajectory files. Input files: http://wyu.dgist.ac.kr/kv12/input/ Trajectory files: http://wyu.dgist.ac.kr/kv12/traj/.

**Funding:** This study was supported by the Creative Research Initiatives of the National Research Foundation (NRF) of Korea (grant number 2008-0061984, http://www.nrf.re.kr/eng/main). Received by Iksoo Chang. The DGIST Core Protein Resources Center funded by MOTIE, Korea (grant number N0001822, http://english.motie.go.kr). Received by Iksoo Chang. The funders had no role in study design, data collection and analysis, decision to publish, or preparation of the manuscript.

**Competing interests:** The authors declare no competing interests.

channel Kv1.2 which senses not only the electrical voltage across the channels but also the cellular acidity. We uncovered that two key residues E327 and H418 in the pore domain of Kv1.2 channel play a role as pH-switch in that their protonation control the gating of the pore in Kv1.2 channel. It offered a molecular insight how the acidity reduces the ion permeability in voltage-gated potassium channels.

## Introduction

Electrical signals in neurons are generated by sequential gating of several voltage-gated ion channels on their cell membranes. The opening and closing of these channels are not only sensitively controlled by membrane potentials in general, but also respond to the intra- and extracellular conditions, such as chemicals [1, 2], mechanical pressure [3], temperature [4], and proton concentrations [5]. Among these channels, the voltage-gated potassium channels (Kv) are selectively permeable to potassium ions and repolarize the membrane potential in response to depolarizing voltage [6]. The molecular mechanisms underlying this potassium ion-selectivity and voltage-dependent gating of the Kv channels have been extensively studied [7–11]. However, the molecular mechanism of pH-dependent gating in Kv channels is less well-understood, although it has been revealed that the potassium ion permeability is inhibited by the high proton concentration in acidosis [12]. The low permeability in the Kv channels is intimately correlated with the prolonged action potential duration observed in acidosis diseases such as cardiac arrhythmias.

These Kv channels have a tetrameric structure composed of four homo-subunits surrounding an ion-transporting pore, with each subunit containing six membrane-spanning α-helices called S1–S6. They are spatially separated from a voltage-sensing domain-containing S1–S4 and a central pore domain-containing S5–S6 that includes a P-helix (Fig 1A). These two domains are connected by an S4–S5 helical linker [13]. The pore domain contains a potassium ion-selective pathway and gates spanning the cell membrane. The narrowest part (extracellular side) on the pore in the channels is the "selectivity filter", whereas the opposite part (intracellular side) of the filter on the pore is the "water-filled cavity" (Fig 1B). The gating of the Kv channels is structurally determined by whether the water-filled cavity is wetted or dewetted.

In Kv1.2 channels, as in the other members of the non-inactivating ion channel family, ionic currents flow in response to the applied depolarized voltage and are maintained until the end of depolarization. Kv1.2 channels maintain the closed conformation at polarized (resting) potentials or the open conformation at depolarized potentials. A permeability of ion channels is defined by the ionic currents per surface area. The appearance of ionic currents is induced by the opening transformation of the channels with wetting water cavity. On the other hand, the disappearance of ionic currents is accompanied by the closing transformation of the channels with dewetting cavity [7]. Previous experimental work at depolarized voltage showed that the permeability of Kv1.2 channels to potassium ions gradually decreases as the pH changes from 7.5 to 4.5 (midpoint at around 5.3), whereas pH-independent zero permeability appeared at resting voltage [12]. This implies that the protonation of some titratable residues make the channels resist the transition from closed to open conformations against depolarized voltage.

Here, we report our observations from an all-atom molecular dynamics (MD) simulation. We found that the voltage-gated Kv1.2 ion channel has dual-functionality due to protonation of the conserved residues E327 and H418 situated near the water cavity on the intracellular side because it induces the gating transition of the pore domain from an open to closed position under acidic conditions. We characterized the structural role of the two determinant

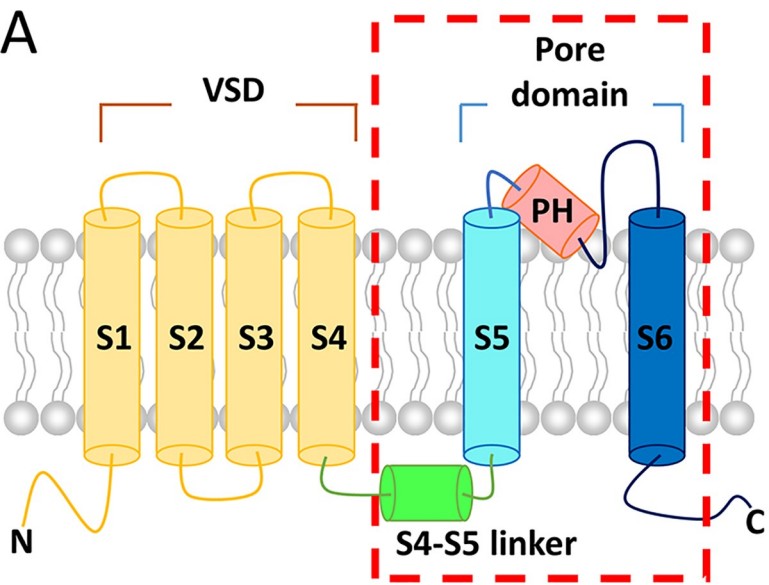

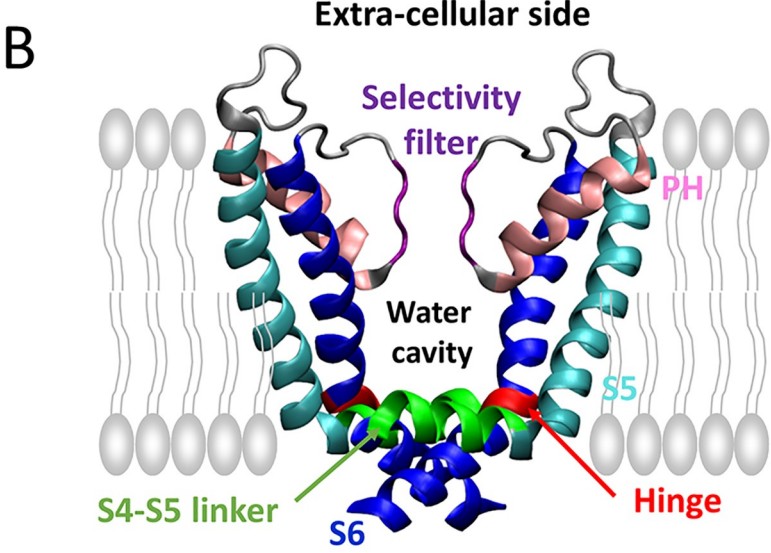

**Fig 1. Schematic illustration of the Kv channel structure. (A)** A cartoon model of a subunit of the Kv channel showing the voltage-sensing domains (VSDs) (S1–S4) in yellow, the S4–S5 helical linker in green, the S5 helix in cyan, the P-helix in pink, and the S6 helix in blue. **(B)** Two opposite subunits of the pore domain of the Kv channel are represented by ribbons. The other two subunits have been omitted. The selectivity filter is shown in purple, and the water cavity is located below the water cavity. The hinge region in S6 helix is highlighted in red.

residues E327 and H418 in the gating of Kv1.2 channels, which can be protonated under a reasonable acidic condition, as demonstrated in the previous experimental study [12]. We suggest the molecular and structural mechanism underlying the acid-induced low permeability of Kv1.2 channels by uncovering the mechanism under acidic conditions.

## Results

### Identification of the key acid-responding residues in Kv1.2

To identify the key residues that might be protonated under acidic pH and resting voltage conditions, that is, from pH 7.5 to around pH 5.3, we attempted to estimate the pKa values of the titratable residues in Kv1.2 channels with closed conformations. The Kv1.2 channel is inactivated with zero permeability in the resting voltage [12]. Thus, physiologically the default state of the Kv1.2 channel is an inactivated state with the pore closure. Nevertheless, the unprotonated wild-type (named Wild$^{UnP}$) conformation provided by PDB 2R9R [14] was in the open conformation. Therefore, we needed to generate the closed structure of Kv1.2 channel from the opened structure either by applying the resting voltage (about -60mV) or the acidic pH to an initial opened structure. We, however, noted that our MD simulation was done only with the pore domain of the channel while the voltage sensing domain (VSD) of the channel is necessary to induce such conformational change subject to the application of the resting voltage. In our setting of MD simulation, we therefore applied the strong acidic condition (namely protonating E327/H418/E420) to an initial opened structure in order to obtain the closed state in the physiological condition for Kv1.2 pore domain. Here, we considered the pore domains (residue numbers: 312–421) that are responsible for the gating of the Kv1.2 channel, excluding the voltage-sensing domains. Based on the representative structural ensembles of the closed conformations in forth trajectory (S5E Fig), the pKa values of the titratable residues in the channel were estimated (Fig 2A) [15]. The pKa values of both E327 in the A- and D-chain and H418 were around 6.0, which means that these two residues can be protonated under the above acidic conditions. Here, pKa of E327 residues were seemed to reside in the 2-fold symmetry whereas pKa of H418 residues remained in the 4-fold symmetry. The closed conformations of the channel, considered for our pKa estimations, were shown to maintain 2-fold symmetry near the water cavity, while the other parts of the channel were remained in 4-fold symmetry (Upper panels of Fig 3A). Presumably, the pKa of E327 residues sensitively reacted to the structural change near the water cavity with 2-fold symmetry rather the pKa of H418 residues. Multiple sequence alignment for Kv1 subfamily proteins showed that almost residues are conserved (Fig 2B). Among these conserved residues two residues E327 and H418 are located near the water-filled cavity, and many charged amino acids are distributed around E327 and H418. Since the protonation of E327 and H418 residues could affect the conformational transition of the pore domain of the Kv1.2 channel, we decided to investigate the effect of protonation of the E327 and H418.

### Protonation of E327 and H418 induces closure of Kv1.2

We performed atomistic MD simulations of the central pore domain in the Kv1.2 channel for both a Ep327/Hp418 state (here, "p" indicates the protonation; details of our MD simulations are provided in the Supplementary Information) and Wild$^{UnP}$ (as a control group). Starting from the initial open conformation of the central pore domain in the Kv1.2 channel, five trajectories of MD simulations for each of a Wild$^{UnP}$ and a Ep327/Hp418 state were run for 2 μs, and the conformational ensembles of the central pore domain in the Kv1.2 channel were sampled (Fig 3A) [16]. The number of water molecules in the cavity implicates that the closed conformations were sampled in Ep327/Hp418 (Fig 3B and S3 Fig), but not in Willd$^{UnP}$ [17]. This result demonstrates that the protonation of E327 and H418 destabilizes the charge interaction network of R326, E327, and H418 under acidic condition. As a result, it induces conformational change from open to close form (Fig 3B and 3C).

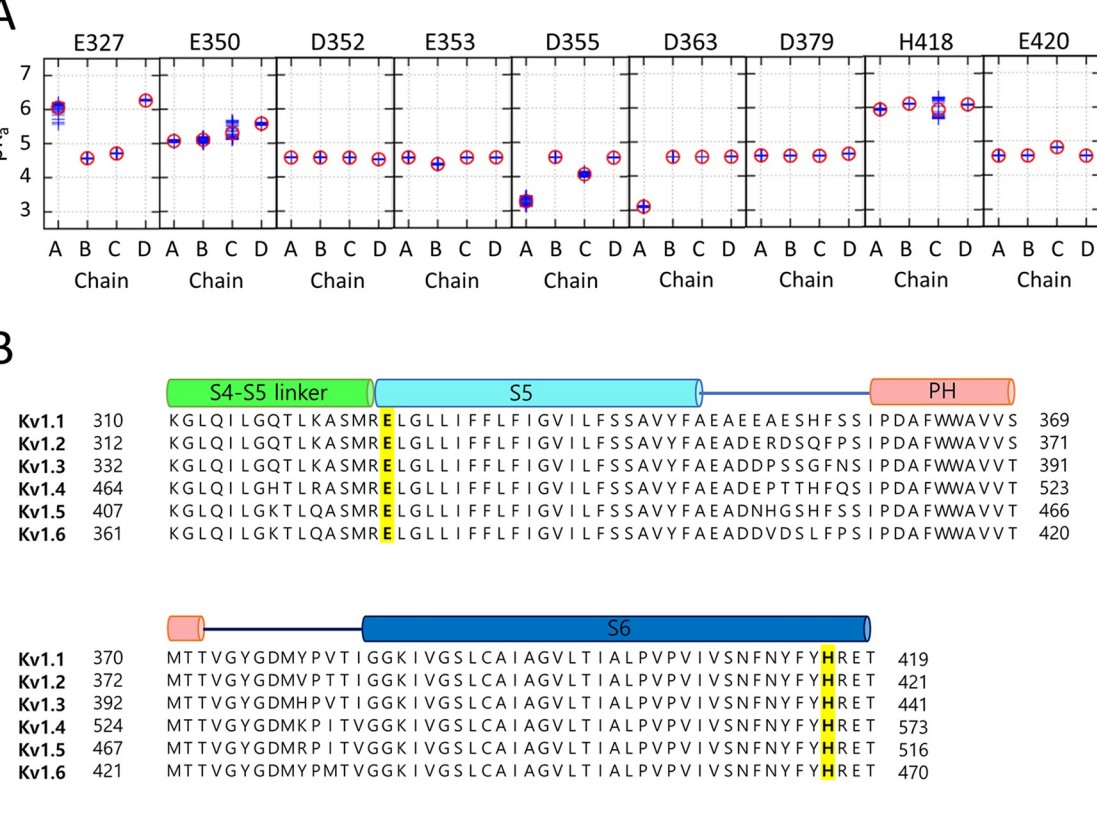

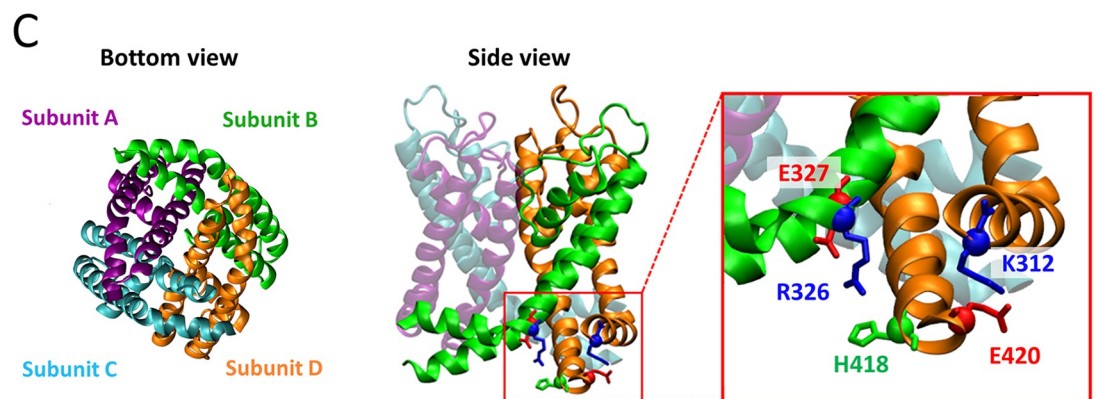

**Fig 2. pKa estimates of H418 and acidic residues and two key residues in the Kv1.2 channel.** (A) The pKa estimates of H418 and the acidic residues in the pore domain of the closed Kv1.2 channels. (B) Multiple sequence alignment for the pore domains of the rat Kv1 subfamily. The cylinders above the sequences denote the secondary structural information of the four helical segments. The two key conserved residues E327 and H418 are highlighted in yellow and bold font. (C) Bottom and side views of the close pore domain of the Kv1.2 channel. The four subunits are represented by different colors in the ribbon diagram. The magnified view shows the key residues E327 and H418 and their neighbors in the inner helical bundle, which are represented by the licorice and Cα balls.

## Structural and molecular mechanisms of the pore closure of Kv1.2

Based on the structural ensembles collected from our MD simulations, we quantified the structural alterations between the open and close form of Kv1.2 channels. The PVP (P405-V406-P407) motif which is located in the middle of H6 (Fig 1B) is flexible and acts as a hinge [17, 18]. This flexible hinge allows the change of kinked angle during channel gating.

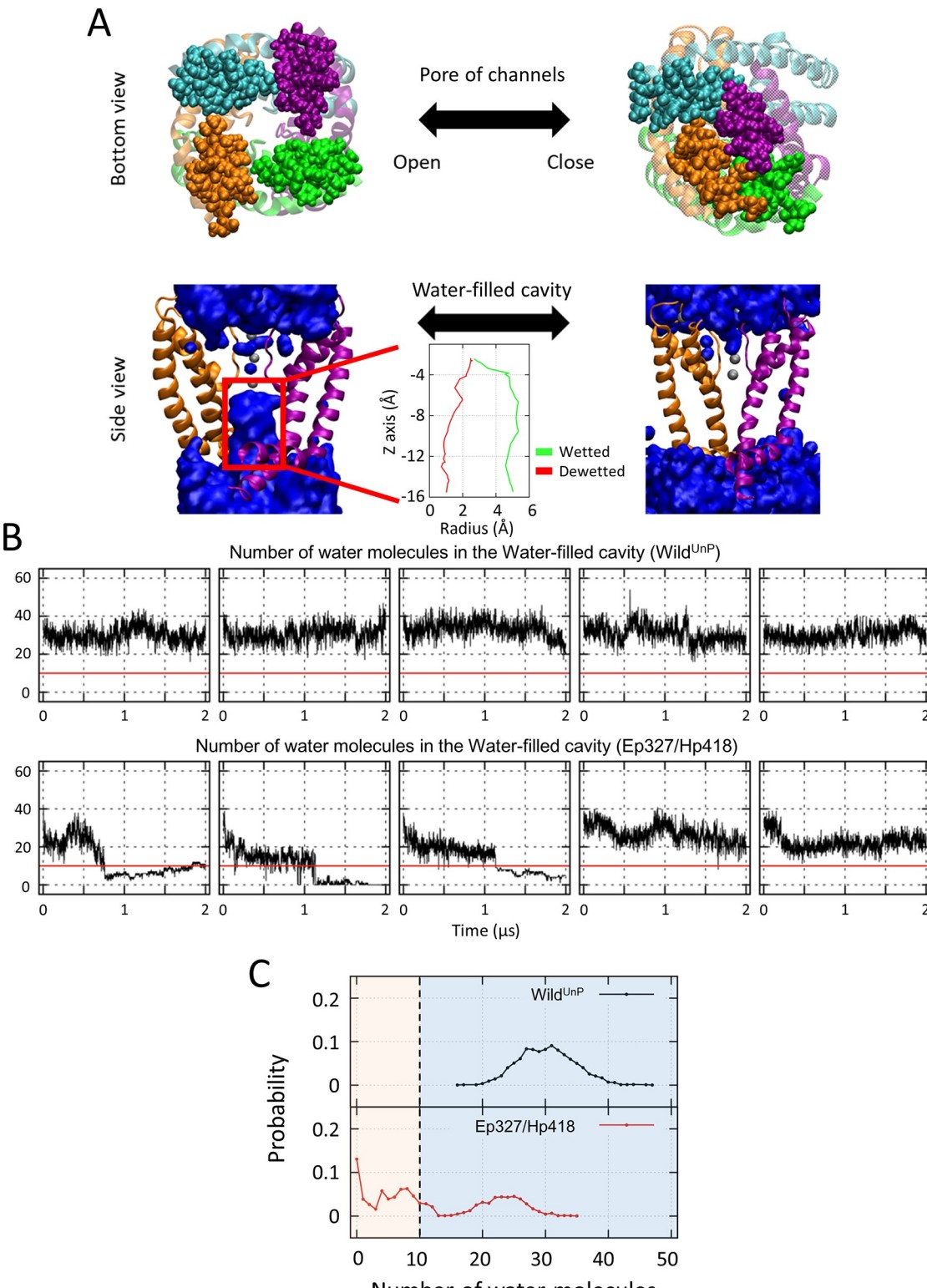

**Fig 3. Pore closure due to protonation of the two key residues E327 and H418 in the Kv1.2 channel. (A)** The structures of the open and closed Kv1.2 channel. Here, the ribbons represent the pore domain of Kv1.2 channels, the blue spheres denote water molecules, and the gray spheres denote potassium ions. The upper panel illustrates the bottom view of the open and closed conformations. The lower panel shows the wetted and dewetted water cavity in the channel. Both the two opposite subunits and lipid molecules have been omitted from the lower panel. **(B)** The time evolution of the number of water molecules in the water

cavity of the Wild$^{UnP}$ or Ep327/Hp418 states of the Kv1.2 channel. The five individual plots in each state represent the simulation results from each trajectory of our MD simulations. In addition, the red horizontal line separates the wetted state from the dewetted state of the water cavity. **(C)** The distributions of the number of water molecules in the water cavity. The left region with fewer water molecules corresponds to the dewetting condition, whereas the right region with many water molecules corresponds to the wetting condition.

We studied how inter-subunit interaction (R326, E327, and H418) affect the channel gating (Fig 4A). We used two structural determinants that can distinguish the open conformation from the closed conformation of the pore in Kv1.2 channels, namely, the R326–H418 inter-subunit distance defined by the nearest inter-atom distances in these two residues (the bottom view in Fig 4A) and the dihedral angle extended by the positions of the Cα atoms in four residues (L393, L400, V408, and Y415) on the S6 helix (yellow ball of the side view in Fig 4A) as a measure of the kinked angle of S6. The dihedral angle determines whether the S6 helix is bent or straight [18].

The Supplementary Information in S1C, S1D, S3C and S3D Figs shows the time evolution of the values of the R326–H418 inter-subunit distance and the dihedral angle from the structural ensembles collected in the last 500 ns time window of our MD simulations, which demonstrates the effects of the protonation of E327 and H418. The R326–H418 distances in the Ep327/Hp418 state of the Kv1.2 channels become longer and the distance distribution is much broader compared with those of the Wild$^{UnP}$, for which the most frequent distances are around 6 Å (Fig 4B). The drastic increase in the R326–H418 distances in the Ep327/Hp418 state is due to the repulsive Coulomb interaction between R326 and Hp418. The distribution of the dihedral angles extended by L393, L400, V408, and Y415 on the S6 helix of the Wild$^{UnP}$ Kv1.2 channels has a distinct peak at around 130˚, indicating that the S6 helix is bent (the dotted black line in Fig 4C). On the other hand, the protonation of both E327 and H418 gives rise to a secondary peak around 245˚ in their distribution, indicating that the S6 helix is straightened (the solid red line in Fig 4C). The increase in the R326–H418 distance is closely correlated with the increase in the dihedral angle extended by L393, L400, V408, and Y415 on the S6 helix and is well captured by the heat map of the ensemble population along the two axes of each quantity (Fig 4D). The heat map in the Ep327/Hp418 state revealed that the increase in the R326–H418 inter-subunit distance straightened the S6 helix. The change in the degree of the dewetting in the cavity of the Kv1.2 channels is also closely correlated with the straightening of the S6 helix. This close correlation is demonstrated in the heat map of the ensemble population along the two axes of the dihedral angle and in the number of water molecules in the cavity (Fig 4E).

Overall, Hp418 was electrostatically pushed by R326 at the end of the S6 helix under acidic conditions. This changed the shape of the S6 helix from bent to straight. Therefore, the straightening of the S6 helix is a robust indication of pore closure in the potassium channel Kv1.2 [19–21]. Here, we suggest that the repulsive Coulomb interaction of the inter-subunits triggered by the protonation of the two key residues E327 and H418 is the molecular mechanism of the pore closure in the Kv1.2 channels, together with both the increase in the inter-subunit distance and the straightening of the S6 helix.

## Altering the conformation of Kv1.2 through protonation or mutation

The conformation of the Kv1.2 channel was further probed by examining the changes in the inter-subunit interactions among R326, E327, and H418 and the intra-subunit interaction between K312 and E420. We were able to modify the charge states of these residues through protonation or mutation. First, we modified the inter-subunit interaction by changing the

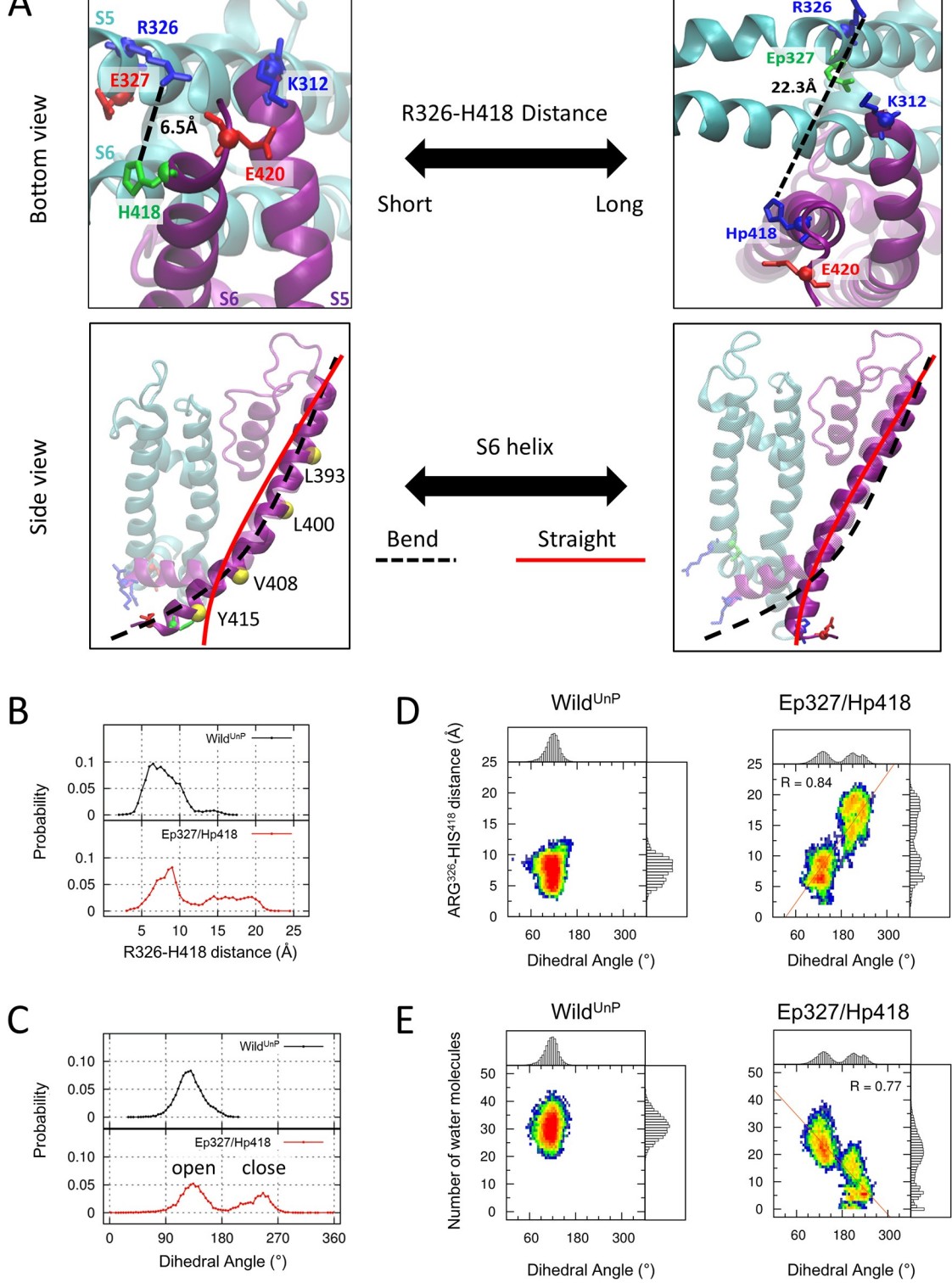

**Fig 4. Straightening of the S6 helix due to protonation of the two key residues E327 and H418 in the Kv1.2 channel. (A)** The detailed structures of the open (left panels) and closed (right panels) Kv1.2 channels. The upper panel illustrates the distance between the R326 and H418 residues. For the open conformation of the pore, the distance between E327 and H418 is 7.6 Å, whereas it is 14.7 Å for the closed conformation. In addition, the lower panel shows the S6 helix when it is bent (black dotted line) or straight (solid red line). Here, two neighbor subunits are displayed using purple and cyan color, respectively, while the other subunits have been omitted. **(B)** The probability distribution curves for the inter-subunit distances between the R326 and H418 residues of the

Wild$^{UnP}$ and Ep327/Hp418 states. **(C)** The probability distribution of the dihedral angles, extended by the position of the Cα atoms in L393, L400, V408, and Y415, for each state. The left half region with the angle smaller than 180˚ corresponds to the bent S6 helix, whereas the right half and the secondary peak is the straight S6 helix. **(D)** Ensembles population of log scale for the R326–H418 distances and the dihedral angles. In the right panel for the Ep327/Hp418 state, a high correlation value of 0.84 was detected. **(E)** Ensembles population of log scale for the dihedral angles and the number of water molecules in the water-filled cavity. In the right panel for the Ep327/Hp418 state, a high correlation value of 0.77 was also detected.

charge state of H418 to Hp418 (S2 Fig) or H418R (S6 Fig) which resembled the weak acidic condition. Second, we broke the inter-subunit interactions by changing to Ep327/Hp418 (S3 Fig) or E327A/H418R (S7 Fig) which resembled the strong acidic condition. Third, we modified the intra-subunit interaction by changing the charge state of E420 to Ep420 (S4 Fig) or E420A (S8 Fig). The protonation or mutation of E420 weakens the charge interaction between E420 and K312. It destabilized the open conformation of the Kv1.2 channel and induced conformational change from open to closed. Additionally, these effect on K312, which located in the S4-S5 linker, could propagate to the VSD throughout the S4 helix. Finally, we broke both the inter- and intra-subunit interactions by changing to Ep327/Hp418/Ep420 (S5 Fig) or E327A/H418R/E420A (S9 Fig) which resembled stronger acidic condition.

Our MD simulations for these 8 variants showed the structural transition from the open to the closed conformation in the pore of the Kv1.2 channel for at least 1 out of 5 trajectories (S2E to S9E Figs). The closed conformations of the variants commonly showed both an increase in the inter-subunit distance (S2C to S9C Figs) and a straightening of the S6 helix (S2D to S9D Figs), as like as the molecular mechanism of the pore closure of Ep327/Hp418 state. Of the variants, the correlation values between the R326-H418 distance and the dihedral angles range from 0.64 to 0.89 (S2A to S9A Figs), and the correlation values between the number of cavity water and the dihedral angles range from 0.49 to 0.78 (S2B to S9B Figs). We were thus able to control the pore closure of Kv1.2 through the various protonation or mutation, which corroborated the molecular mechanism regarding the effect of charge-charge interactions on the structure and gating of the pore domain of Kv1.2 channel.

## Discussion

We investigated the molecular mechanism underlying the pH-dependent gating of the pore domain of the Kv1.2 channel protein under intracellular acidic conditions. A decrease in environmental pH from 6 to 5 with depolarized voltage causes Kv1.2 to undergo a conformational change from open to closed [12]. Our pKa estimates indicate that only two amino acids E327 and H418 change their charge states in response to a change in the environmental pH. Thus, E327 and H418 are proposed as key residues for pH sensing. To assess the role of the key residues under acidic pH conditions, we performed MD simulations with key protonated residues. Ep327 and Hp418 highly destabilize the electrostatic interaction. Inter- and intra-subunit interactions, which consists of K312, R326, E327, H418, and E420, were critically destabilized by the change in the charge states of these titratable residues. Because the net charge of this cluster changes from zero to positive (+1, +2, or +3), the intra-subunit interaction K312-Ep420 weakens and the repulsive force between R326 and Hp418 increases the distance. This repulsive force pushes the end of S6, leading to its distortion. In our simulation, the channels are perfectly closed when the two opposite subunits of S6 undergo a conformational change. Two distorted S6 helices move close together and fill the space previously occupied by the water. As a result, the closed conformation of Kv1.2 under acidic conditions was induced by the protonated E327 and H418. We tracked the step-by-step conformational change using MD simulation. H418R and E327A/H418R mutants undergo a structural change from an open to closed conformation. Thus, we were able to modulate the conformation of Kv1.2 via

in silico mutation of the key residues. It imply that H418R and E327A/H418R mutants have low ion permeability, even under the neutral pH condition. With various protonated or mutated MD simulations, we found that the charge interaction of inter- and intra-subunit affects the channel gating. However, because of the sampling issues, we are not sure about a quantitative comparison that which interactions are more important. Therefore, additional in-vivo or in-vitro experiments are needed for the quantitative comparison.

The molecular mechanism of the voltage-dependent gating of the Kv channels involves displacement of the VSD that regulate the wetting or dewetting of the cavity [7]. The previous study shows that the S4 helix is the main moving part in the VSD which moved down ~15Å across the membrane during the deactivation, and the motion of S4 affects the pore domain through the S4-S5 linker [7]. In our simulation, the intra-subunit interaction between K312 and E420 is affected by the protonation state of E420, so the K312 (locate at the S4-S5 linker) could restrict the motion of S4 depending on the environmental pH condition. As a result, S4-S5 linker is affected by environmental pH. Since we simulated with the pore domain (residues 312–421, without VSDs) of Kv1.2, it does not fully reflect the realistic motion of a full chain of Kv1.2 with VSDs. However, it would be used to understand the effect of the acidic environment on channel gating of the pore domain. The coexistence of these voltage- and acid-dependent gating mechanisms in Kv1.2 channels implies that both the voltage-induced structural pressure and acid-induced structural pressure can simultaneously influence the gating of the channels. The possibility for this simultaneous action of two different mechanisms of pore gating was indicated in previous experimental studies of the behavior of the potassium ion current of Kv1.2 channels that suggested that the acidity competes with the depolarizing voltage [12]. At a glance, this might be counterintuitive because the neuronal channels were not able to distinguish the environmental factor that played a role in their own gating. However, it is worth noting that the time scale for the fluctuation in the membrane potential is on the order of milliseconds, whereas that of the cellular pH is from seconds to minutes. Thus, the permeability of potassium ion currents through Kv1.2 channels is not only finely controlled by the depolarizing voltage in the short time interval but also governed by the acidity in the long time interval. Our study offers insight into the dual-gating mechanisms of the Kv channels, which orchestrate both the voltage-dependent and pH-dependent gating mechanisms of different molecular mechanisms.

## Methods

### All-atom molecular dynamics simulation

We performed MD simulations using GROMACS 5.0 [22] with the CHARMM36 force field [23]. The initial configurations of the Kv1.2 channel for the MD simulation were generated using CHARMM-GUI lipid builder [24]. The system consists of a channel protein, lipids (271 POPE in upper and lower leaflets), water molecules (~10,000 TIP3P water molecules). In order to conduct the simulation under the same ionic conditions in the previous studies [7, 19], we added 0.6M of $K^+$ and $Cl^-$ ions. The channel protein is a symmetric tetramer structure with an open-pore domain (residues 312–421) that is derived from the X-ray crystal structure of the Kv1.2 channel (PDB code: 2R9R) [14]. The system includes ~71,000 atoms in a rectangular box ($100 \times 100 \times 75$ Å) under the periodic boundary condition. The particle-mesh Ewald (PME) method [25] was applied for assessing long-range electrostatic interactions with a 12 Å cut-off distance, and potential-based switching functions were used for van der Waals interactions with a 10–12 Å switching range. The position of the hydrogen atoms was restrained by the equilibrium bond length using the LINCS algorithm [26]. Approximately 5,000 steps of steepest-descent minimization were conducted. We performed the heating process by

gradually removing the restraint on lipids and protein over 20,000 steps for the stable 310K system temperature. An additional 30 ns simulation with a restrained protein backbone was performed to equilibrate the water cavity position, followed by a production run. The production simulations were carried out for 2 μs with a 2 fs time step in NPT ensemble holding a constant particle number (N = ~71,000), pressure (P = 1 bar), and temperature (T = 310 K). Temperature was controlled by the Nosé–Hoover temperature coupling method [27, 28] with a tau-t of 1 ps and pressure was maintained by the semi-isotropic Parrinello–Rahman method [29, 30] with a tau-p of 5 ps and compressibility of $4.5 \times 10^{-5}$ bar$^{-1}$. All trajectories were recorded every 10 ps, and VMD software [31] was used for the visual analysis.

## pKa calculation

For pKa estimates of H418 and the acidic residues in the pore domain of the closed Kv1.2 channels, several processes were executed. 1) We extracted 98 ensemble structures of the Kv1.2 channels with pore closure in the last 1 μs time window from our MD simulation for an Ep327/Hp418/Ep420 state. In our setting of MD simulation the pore domain of Kv1.2 channel together with explicit membrane lipids, water molecules, and ions were included. 2) In order to select preferred titration states among all possible titration states in the pH range from 3 to 8, we took advantage of MEAD algorithms [15] to the 10 out of 98 pore closure ensembles with these options (dielectric constants of a molecular interior region / solvent region were $\varepsilon_{in}$ = 6.0, $\varepsilon_{sol}$ = 80.0, ionic strength was 0.15 mol/l, and the effect of membrane was excluded.). 3) We selected 765 titration states of the channels that were predominant in the pH range from 3 to 8. For the estimation of pKa values we utilized these 765 titration states and 10 pore closure structural ensembles from our MD simulation. 4) At each titration state on all 98 ensemble structures, we calculated the system energies in the implicit water using AMBER force field 99SB [32]. 5) From the titration states of 3) and the system energies of 4), we reconstructed the partition function, the protonation fractions and estimated pKa values of H418 and the acidic residues (Fig 2) [33]. Red circles in the figure represent the pKa values when the average energies (using AMBER force field 99SB) of ensemble structures, $\langle E \rangle$, were used to calculate the protonated fraction as a function of pH for the corresponding titration states. The blue crosses represent the pK$_a$ values when the statistical variance of the energies $\langle E \rangle \pm 0.1\sigma, \langle E \rangle \pm 0.2\sigma, ..., \langle E \rangle \pm 1\sigma$ was considered to reflect the uncertainty of the pK$_a$ values due to the structural fluctuation of ensemble structures.

## Supporting information

**S1 Fig. Simulation results for Wild$^{UnP}$. (A)** Ensembles population of log for the R326–H418 distances and dihedral angles, and **(B)** ensembles population of log scale for the dihedral angles and number of water molecules in the water-filled cavity. The time evolution of **(C)** the distance between R326 and H418 residues, **(D)** the dihedral angle, and **(E)** the number of water molecules in the water cavity from five individual trajectories for Wild$^{UnP}$.
(TIF)

**S2 Fig. Simulation results for Hp418. (A)** Ensembles population of log for the R326–H418 distances and dihedral angles, and **(B)** ensembles population of log scale for the dihedral angles and number of water molecules in the water-filled cavity. The time evolution of **(C)** the distance between R326 and H418 residues, **(D)** the dihedral angle, and **(E)** the number of water molecules in the water cavity from five individual trajectories for Hp418.
(TIF)

**S3 Fig. Simulation results for Ep327/Hp418. (A)** Ensembles population of log for the R326–H418 distances and dihedral angles, and **(B)** ensembles population of log scale for the dihedral angles and number of water molecules in the water-filled cavity. The time evolution of **(C)** the distance between R326 and H418 residues, **(D)** the dihedral angle, and **(E)** the number of water molecules in the water cavity from five individual trajectories for Ep327/Hp418.
(TIF)

**S4 Fig. Simulation results for Ep420. (A)** Ensembles population of log for the R326–H418 distances and dihedral angles, and **(B)** ensembles population of log scale for the dihedral angles and number of water molecules in the water-filled cavity. The time evolution of **(C)** the distance between R326 and H418 residues, **(D)** the dihedral angle, and **(E)** the number of water molecules in the water cavity from five individual trajectories for Ep420.
(TIF)

**S5 Fig. Simulation results for Ep327/Hp418/Ep420. (A)** Ensembles population of log for the R326–H418 distances and dihedral angles, and **(B)** ensembles population of log scale for the dihedral angles and number of water molecules in the water-filled cavity. The time evolution of **(C)** the distance between R326 and H418 residues, **(D)** the dihedral angle, and **(E)** the number of water molecules in the water cavity from five individual trajectories for Ep327/Hp418/Ep420.
(TIF)

**S6 Fig. Simulation results for H418R. (A)** Ensembles population of log for the R326–H418 distances and dihedral angles, and **(B)** ensembles population of log scale for the dihedral angles and number of water molecules in the water-filled cavity. The time evolution of **(C)** the distance between R326 and H418 residues, **(D)** the dihedral angle, and **(E)** the number of water molecules in the water cavity from five individual trajectories for H418R.
(TIF)

**S7 Fig. Simulation results for E327A/H418R. (A)** Ensembles population of log for the R326–H418 distances and dihedral angles, and **(B)** ensembles population of log scale for the dihedral angles and number of water molecules in the water-filled cavity. The time evolution of **(C)** the distance between R326 and H418 residues, **(D)** the dihedral angle, and **(E)** the number of water molecules in the water cavity from five individual trajectories for E327A/H418R.
(TIF)

**S8 Fig. Simulation results for E420A. (A)** Ensembles population of log for the R326–H418 distances and dihedral angles, and **(B)** ensembles population of log scale for the dihedral angles and number of water molecules in the water-filled cavity. The time evolution of **(C)** the distance between R326 and H418 residues, **(D)** the dihedral angle, and **(E)** the number of water molecules in the water cavity from five individual trajectories for E420A.
(TIF)

**S9 Fig. Simulation results for E327A/H418R/E420A. (A)** Ensembles population of log for the R326–H418 distances and dihedral angles, and **(B)** ensembles population of log scale for the dihedral angles and number of water molecules in the water-filled cavity. The time evolution of **(C)** the distance between R326 and H418 residues, **(D)** the dihedral angle, and **(E)** the number of water molecules in the water cavity from five individual trajectories for E327A/H418R/E420A.
(TIF)

## Acknowledgments

We acknowledge DGIST Supercomputing Bigdata Center for the allocation of dedicated supercomputing time. Protein structure images were made with VMD software support. VMD is developed with NIH support by the Theoretical and Computational Biophysics group at the Beckman Institute, University of Illinois at Urbana-Champaign.

## Author Contributions

**Conceptualization:** Juhwan Lee, Mooseok Kang, Sangyeol Kim, Iksoo Chang.

**Data curation:** Juhwan Lee, Sangyeol Kim.

**Formal analysis:** Juhwan Lee, Sangyeol Kim.

**Funding acquisition:** Iksoo Chang.

**Methodology:** Juhwan Lee.

**Project administration:** Iksoo Chang.

**Software:** Mooseok Kang.

**Supervision:** Iksoo Chang.

**Validation:** Juhwan Lee, Sangyeol Kim.

**Visualization:** Juhwan Lee, Sangyeol Kim.

**Writing – original draft:** Juhwan Lee, Mooseok Kang, Sangyeol Kim, Iksoo Chang.

**Writing – review & editing:** Juhwan Lee, Mooseok Kang, Sangyeol Kim, Iksoo Chang.

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
