## [Decision Letter · Decision Letter 0]

8 Nov 2019

Dear Dr Chang,

Thank you very much for submitting your manuscript 'Structural and molecular insight into the pH-induced low-permeability of the voltage-gated potassium channel Kv1.2 through dewetting of the water cavity' for review by PLOS Computational Biology. Your manuscript has been fully evaluated by the PLOS Computational Biology editorial team and in this case also by independent peer reviewers. The reviewers appreciated the attention to an important problem, but raised some substantial concerns about the manuscript as it currently stands. While your manuscript cannot be accepted in its present form, we are willing to consider a revised version in which the issues raised by the reviewers have been adequately addressed. We cannot, of course, promise publication at that time.

Sincerely,

Peter M Kasson

Associate Editor

PLOS Computational Biology

Nir Ben-Tal

Deputy Editor

PLOS Computational Biology

[LINK]

Reviewer's Responses to Questions

**Comments to the Authors:**

Reviewer #1: The authors investigate the pH-dependent gating mechanism for the pore domain of Kv1.2 channel. The authors report that a decrease in pH, leads to protonation of residues H418 and E327. The resulting repulsive coulombic interactions between R326 and Hp418 straightens the S6 helices, leading to de-wetting of the internal cavity and channel closure. These findings could help to better understand the role of pH in the gating mechanism of channels from the Kv family in health and disease conditions.

I would like the authors to address the following comments:

*The authors refer to the deprotonated and protonated states as wild-type and Ep327/Hp418, respectively. This naming should be corrected. Both, the deprotonated and protonated states are “wild-type” if no aminoacid-residue substitution was performed.

*On page 15, the authors mention E327/H418R variant, but according to table 1, it should be E327A/H418R double-mutant.

*The authors should make clear their definition of permeability. From the paper, I understand that it is based on the access of water molecules to the internal cavity of the channel. However, it can be confused with ion permeation events, for which the authors didn’t report any occurrence.

*The authors claim to have modified “genetically” the permeability of Kv1.2 via mutation of key residues (page 16). This can be understood as if the authors performed experimental mutagenesis, which they didn’t. Hence, I would recommend using “in silico” instead of “genetically” in this sentence to avoid confusion.

*The authors should discuss if pH changes not only affect gating by leading to closure of the pore domain as they propose. For instance should pH-decrease also affect voltage sensing and gating through modification of protonation states of key residues at the voltage-sensor domain of this channel?

Regarding methodologies:

*It is not clear why the authors used the Ep327/Hp418/Ep420 MD simulation for pKa estimates. Why not to use the MD simulation of the wild-type instead?

*Could the authors discuss why the pKa analysis shows that all four H418 residues could be protonated but only two of the four E327 residues?

*The statistical significance of the results is not clear. For instance, is 3/5 closing events statistically different from 2/5, 1/5 or 0/5 closing events? The authors should increase the number of replicas per system and use some statistical figure or metric for comparison.

*For each system, were the replicas run from different initial conditions?

Reviewer #2: Review of PCOMPBIOL-D-19-01546

This is a potentially interesting computational study of pH-induced low-permeability of Kv1.2 channels. The authors suggest on the basis of simulations that acid-induced reduction in permeability of this channel is due to protonation of E327 and H148 which in turn leads to loss of curvature/bending of the helices and a consequent dewetting of water-filled cavity. This would be an interesting result. However I have some questions about the underlying logic of how the simulations were designed and performed.

The pKa calculations use MEAD – does this work for membrane proteins? How is the bilayer modelled (e.g. as a region of low dielectric)? This needs to be made explicit. The description in the Methods (page 17) is very unclear – I am still uncertain as to what was actually done even after reading is several times. It also fails to mention that the use of red circles and blue crosses refers to Fig. 2A which does not help.

From the main Results text (page 12) and Fig. 2 it is stated that “pKa values of both E327 in the A- and D-chain and H418 were around 6.0”. From Fig. 2A it can be seen that the values for E327 of the B & C chains are ~4.5. To me this implies there must have been loss of 4-fold symmetry in the closed state model – this should be described and discussed.

After describing the pKas for the closed state (model) they then go back to the open state structure (page 13), protonate E327 and H418 (earlier on they also protonated E420, page 12) and show that the channel closes. What is the relationship of the Ep327/Hp418 closed state model to the previous Ep327/Hp418/Ep420 model? Are they the same? Do they show comparable breakdown of 4-fold symmetry.

This then leads me to question the logic of the design of the simulations. Would it not be more logical to start with the (experimental) open state, calculate pKas, decide on a protonation state for lowered pH and then run the simulations to see if the channel closes? This seems to me to be a major flaw in the design of the study, and needs to be addressed directly by running the open state pKa calculations and simulations based on these (or at least explaining the logic of the simulation design used).

I have a couple of more technical points:

Fig. 3 – This describes pore closure. It would be helpful to have pore radius profiles shown for that start and end of each simulation. These should reveal to what extent dewetting is the consequence of pore narrowing/occlusion.

Fig. 4 – I suggest better descriptions of the S6 helix would be ‘bent’ and ‘straight’. However, it looks to me that even after ‘straightening’ the S6 helix remains kinked, but in a different direction. Helix kink angles can be quantified and perhaps this is needed to assess the reproducibility of the open -> closed transition in the various simulations.

Reviewer #3: This manuscript presents a molecular dynamics study of voltage-gated K+ channel Kv1.2. The authors investigate the effect of protonating residues located on the intracellular side of the channel on channel closure. Protonating two residues, E327 and H418, results in the loss of electrostatic interactions at the end of the S6 helix and correlate with S6 straightening and pore dewetting, providing molecular insight into the observed reduction in ion current at low pH.

The paper is interesting and the work appears to be well executed, but clarifying the rationale for the calculations and extending the analysis of the results presented to all the systems studied would strengthen the paper. In particular:

1. The rationale for the simulations is unclear. Why was only the closed state of the channel considered for pKa calculations? If conduction is lower but not abolished at low pH, doesn't that suggest that the channel still has a significant propensity to be open?

2. Why were only two out of four copies of E327 estimated to have a higher pKa? On the one hand, this result indicates that the simulations have not converged (the four residues should have identical pKa). On the other hand, if taken at face value, this result questions the rationale for protonating all four E327 residues in the subsequent simulations.

3. Most importantly, the manuscript indicates that as many as 8 variants of the channel were simulated (Table 1), but detailed analysis is only reported for 4 of them (Ep/Hp, EA/HR, HR, and Ep/Hp/Ep), and, surprisingly, a full analysis of the results is only provided for the Ep/Hp system without any justification for neglecting the other systems. In particular, full analysis of the above as well as the other channel variants in which dewetting was observed (namely, Ep, Hp, EA and EA/HR/EA) should be provided in order to test the validity and the generality of the mechanism proposed by the authors, linking the changes in electrostatic interactions to helix straightening and pore dewetting.

In other words, limiting the full analysis to doubly-protonated E327 and H418 is not justified. Since all the other variants lead to pore dewetting, they should all be analyzed to the same extent as the wild type and Ep/Hp variants.

4. Contrary to the claim made by the authors, the multiple sequence alignment presented in Fig. 2B does not provide rigorous evidence that the 2 residues considered are highly conserved, since most residues are conserved in the sequences shown.

5. It should be noted that the simulations have not reached equilibrium, since some trajectories led to dewetting while some others remained wetted. As such, the simulations do not provide a rigorous estimate of the relative stability of wetted and dewetted states. Although the fact that dewetting is observed in the doubly-protonated system and not in the wild type suggests that dewetting is favoured by double protonation, strictly speaking it does not rule out that dewetting could occur in the wild type over longer time scales—in other words, the apparent relative kinetic stability (or lack thereof) of the wetted state does not imply its thermodynamic stability (or lack thereof). Therefore, statements such as “This result demonstrates that the pore domain of Kv1.2 channels prefers the closed conformations under acidic conditions”, as well as multiple other statements regarding such “preference”, are not supported by the results provided. The text should be carefully revised to omit any implication that the simulations have reached equilibrium.

6. The text states: “The hinge of the S6 helix maintains electrostatic balance through two inter-subunit interactions of R326-H418 and E327-H418. These interactions stabilize the open conformation of the Kv1.2 pore domain under a neutral pH condition.” The “hinge” of S6 is not define. Moreover, the authors do not provide evidence that these are the only interactions that change upon closing of the channel. Finally, as noted in comment #3 above, the authors should provde corroborating evidence for this mechanism by analyzing the other systems that they simulated as positive controls.

In addition, note that in the interpretation of the results, correlation does not imply causality.

7. What do the authors mean by “Hp418 was electrostatically pushed by R326 at the hinge of the S6 helix”? The helical hinge seems to be far from the residues in question and is not defined (see comment #6 above).

8. The fact that the VSDs are missing from the simulations suggests that the results presented may not have been observed if the VSDs had been present. In other words, the present simulations may not provide a realistic model of the closed state of the full channel. The authors should comment on that in the manuscript.

9. Correlation coefficients should be provided for the left panels of Fig. 5.

10. Methods: Why the very high 0.6 M salt concentration? What did the “20,000 steps of equilibrating simulations” consist of?

11. “After applying MEAD programs” is vague. Enough details should be provided on the pKa calculations for a competent researcher to be able to reproduce the results provided.

12. In the description of pKa estimates, “uncertainty” should be used rather than “flexibility”.

**Have all data underlying the figures and results presented in the manuscript been provided?**

Reviewer #1: Yes

Reviewer #2: No:

Reviewer #3: None

PLOS authors have the option to publish the peer review history of their article (what does this mean?). If published, this will include your full peer review and any attached files.

Reviewer #1: No

Reviewer #2: No

Reviewer #3: No

---

## [Decision Letter · Decision Letter 1]

12 Feb 2020

Dear Prof. Chang,

Thank you very much for submitting your manuscript "Structural and molecular insight into the pH-induced low-permeability of the voltage-gated potassium channel Kv1.2 through dewetting of the water cavity" for consideration at PLOS Computational Biology. As with all papers reviewed by the journal, your manuscript was reviewed by members of the editorial board and by several independent reviewers. The reviewers appreciated the attention to an important topic. Based on the reviews, we are likely to accept this manuscript for publication, providing that you modify the manuscript according to the review recommendations.  While some of the reviewers' remaining changes require attention, we anticipate that this should be feasible.

Sincerely,

Peter M Kasson

Associate Editor

PLOS Computational Biology

Nir Ben-Tal

Deputy Editor

PLOS Computational Biology

[LINK]

Reviewer's Responses to Questions

**Comments to the Authors:**

Reviewer #1: The revised version improved and a lot. I am very pleased to recommend it for publication pending proof-reading and editorial changes. Several of the figures are very complex and multi-panel with low resolution (this could be an artefact of the submission system). I also had difficulties in places to comprehend take-home message due to very convoluted and very long paragraphs. Nonetheless, the story is interesting and will without doubts find its readership.

Reviewer #3: Some of my earlier concerns have not been addressed satisfactorily.

1. In particular, my earlier point #3: To support the conclusions of the paper, it is essential to report the full detailed analysis of all 8 systems studied in order to provide controls for the proposed mechanism. To address this request, the authors have added supplementary figures showing full analysis of the other systems. However, they do not mention or discuss these results explicitly in the main text. This is all the more surprising because the results actually corroborate their mechanistic conclusions regarding the effect of charge-charge interactions on the structure and gating of the pore domain. The authors should discuss these results, which would significantly strengthen the paper.

2. All the responses given to the reviewers should also be reflected in the text. However, this was not done for my previous request #10: the authors should explain the choice of 0.6 M salt concentration not only in their reply to me, but also in the text of the manuscript. This comment also applies to some of the responses made to the other 2 reviewers.

Problems with the interpretation of the results:

3. Since none of the closing simulations are at equilibrium, the quantities reported in the Figures as “free energy landscapes (in arbitrary units)” are wrong on two counts: (a) these are not free energies; and (b) free energy is not unitless. I would recommend that the authors report the results as 2D histograms to avoid these issues.

4. In addition, since none of the closing simulations are at equilibrium (observed dewetting transitions being irreversible), the amount of time spent in the closed state (reported as number of snapshots in Table 1) is meaningless. Whether or not closing occurred early or late in any simulation is essentially random (as it would be in an exponential relaxation process). As such, it is more appropriate to report, as the authors did in the previous version of the paper, the number of simulations that underwent irreversible closing events (i.e., at least 1 out of 5 for all the systems except the unprotonated WT), as an indication of the capacity of the systems to undergo dewetting. While all the systems other than the unprotonated WT underwent closing, based on the data provided and the above considerations, it is not possible to quantify the likelihood that any of these systems would close. As such, it is not possible to rank them either. In other words, ALL THE VARIANTS PRODUCED INDISTINGUISHABLE RESULTS. This point should be conveyed in the text.

5. The simulations show that the channel closes when the total charge of the titratable cluster changes by +1, +2, or +3. However, the authors confine their analysis and conclusions to the effect of a +2 change, when in fact the same result is observed with +1 and +3 changes, both with protonation and with mutations. This should be conveyed in the text.

6. The statement that “the H418 residue is a more suitable target for a single mutation than E327” is not supported by the data shown and should be removed. There is no discernible effect of changing one residue or the other or both.

**Have all data underlying the figures and results presented in the manuscript been provided?**

Reviewer #1: Yes

Reviewer #3: None

PLOS authors have the option to publish the peer review history of their article (what does this mean?). If published, this will include your full peer review and any attached files.

Reviewer #1: No

Reviewer #3: No
---

## [Editor Report · Decision Letter 2]

13 Mar 2020

Dear Prof. Chang,

We are pleased to inform you that your manuscript 'Structural and molecular insight into the pH-induced low-permeability of the voltage-gated potassium channel Kv1.2 through dewetting of the water cavity' has been provisionally accepted for publication in PLOS Computational Biology.

***The editor would also suggest the following minor change to the text***

In your revision, you altered text to read "Ensembles population of log scale".  I would recommend writing "log probabilities of sampled conformations", which I believe more precisely describes the plotted data.

Best regards,

Peter M Kasson

Associate Editor

PLOS Computational Biology

Nir Ben-Tal

Deputy Editor

PLOS Computational Biology

---

## [Editor Report · Acceptance letter]

2 Apr 2020

PCOMPBIOL-D-19-01546R2 

Structural and molecular insight into the pH-induced low-permeability of the voltage-gated potassium channel Kv1.2 through dewetting of the water cavity

Dear Dr Chang,

I am pleased to inform you that your manuscript has been formally accepted for publication in PLOS Computational Biology. Your manuscript is now with our production department and you will be notified of the publication date in due course.

With kind regards,

Laura Mallard
